# Coded Residual Transform for Generalizable Deep Metric Learning

**Shichao Kan[1], Yixiong Liang[1], Min Li[1], Yigang Cen[2,3,*], Jianxin Wang[1], Zhihai He[4,5,*]**

[1]School of Computer Science and Engineering, Central South University, Changsha, Hunan, 410083

[2]Institute of Information Science, School of Computer and Information Technology,
Beijing Jiaotong University, Beijing 100044, China

[3]Beijing Key Laboratory of Advanced Information Science and Network Technology, Beijing 100044, China

[4]Department of Electrical and Electronic Engineering, Southern University of Science and Technology,
Shenzhen, China

[5]Pengcheng Lab, Shenzhen, 518066, China

`kanshichao@csu.edu.cn, yxliang@csu.edu.cn, limin@mail.csu.edu.cn`
**`ygcen@bjtu.edu.cn`**`, jxwang@mail.csu.edu.cn,` **`hezh@sustech.edu.cn`**

## Abstract

A fundamental challenge in deep metric learning is the generalization capability of the feature embedding network model since the embedding network learned on training classes need to be evaluated on new test classes. To address this challenge, in this paper, we introduce a new method called *coded residual transform* (CRT) for deep metric learning to significantly improve its generalization capability. Specifically, we learn a set of diversified prototype features, project the feature map onto each prototype, and then encode its features using their projection residuals weighted by their correlation coefficients with each prototype. The proposed CRT method has the following two unique characteristics. First, it represents and encodes the feature map from a set of complimentary perspectives based on projections onto diversified prototypes. Second, unlike existing transformer-based feature representation approaches which encode the original values of features based on global correlation analysis, the proposed coded residual transform encodes the relative differences between the original features and their projected prototypes. Embedding space density and spectral decay analysis show that this multi-perspective projection onto diversified prototypes and coded residual representation are able to achieve significantly improved generalization capability in metric learning. Finally, to further enhance the generalization performance, we propose to enforce the consistency on their feature similarity matrices between coded residual transforms with different sizes of projection prototypes and embedding dimensions. Our extensive experimental results and ablation studies demonstrate that the proposed CRT method outperform the state-of-the-art deep metric learning methods by large margins and improving upon the current best method by up to 4.28% on the CUB dataset.

## 1 Introduction

Deep metric learning (DML) aims to learn effective features to characterize or represent images, which has important applications in image retrieval [1, 2], image recognition [3], person re-identification [4], image segmentation [5], and tracking [6]. Successful metric learning needs to achieve the following two objectives: (1) *Discriminative*. In the embedded feature space, image features with the same

---

*Corresponding authors

36th Conference on Neural Information Processing Systems (NeurIPS 2022).

semantic labels should be aggregated into compact clusters in the high-dimensional feature space while those from different classes should be well separated from each other. (2) *Generalizable*. The learned features should be able to generalize well from the training images to test images of new classes which have not been seen before. During the past a few years, methods based on deep neural networks, such as metric loss functions design [7, 8, 9, 10, 11], embedding transfer [12, 13, 14, 15, 16], structural matching [17], graph neural networks [2, 18], language guidance [19], and vision transformer [1, 20, 21], have achieved remarkable progress on learning discriminative features. However, the generalization onto unseen new classes remains a significant challenge for existing deep metric learning methods.

In the literature, to improve the deep metric learning performance and alleviate the generalization problem on unseen classes, regularization techniques [22, 15], language-guided DML [19], and feature fusion [2, 8, 18] methods have been developed. Existing approaches to addressing the generalization challenge in metric learning focus on the robustness of linear or kernel-based distance metrics [23, 24], analysis of error bounds of the generalization process [25], and correlation analysis between generalization and structure characteristics of the learned embedding space [22]. It should be noted that in existing methods, the input image is analyzed and transformed as a whole into an embedded feature. In other words, the image is represented and projected globally from a single perspective. We recognize that this single-perspective projection is not able to represent and encode the highly complex and dynamic correlation structures in the high-dimensional feature space since they are being collapsed and globally projected onto one single perspective and the local correlation dynamics have been suppressed. According to our experiments, this single-perspective global projection will increase the marginal variance [26] and consequently degrade the generalization capability of the deep metric learning method.

Furthermore, we observe that existing deep metric learning methods attempt to transform and encode the original features. From the generalization point of view, we find that it is more effective to learn the embedding based on relative difference between features since the absolute value of features may vary significantly from the training to new test classes, but the relative change patterns between features may remain largely invariant. To further understand this idea, consider the following toy example: a face in the daytime may appear much different from a face in the night time due to changes in lighting conditions. However, an effective face detection with sufficient generalization power will not focus on the absolute pixel values of the face image. Instead, it detects the face based on the relative change patterns between neighboring regions inside the face image. Motivated by this observation, in this work, to address the generalization challenge, we propose to learn the embedded feature from the projection residuals of the feature map, instead of its absolute features.

The above two ideas, namely, multi-perspective projection and residual encoding, lead to our proposed method of coded residual transform for deep metric learning. Specifically, we propose to learn a set of diversified prototype features, project the features onto every prototype, and then encode the features using the projection residuals weighted by their correlation coefficients with the target prototype. Unlike existing transformer-based feature representation approaches which encode the original values of features based on global correlation analysis [27, 28, 29], the proposed coded residual transform encodes the relative differences between original features and their projected prototypes. Our extensive experimental results and ablation studies demonstrate that the proposed CRT method is able to improve the generalization performance of deep metric learning, outperforming the state-of-the-art methods by large margins and improving upon the current best method by up to 4.28%.

We learn those projection prototypes based on the training classes and transfer them into the test classes. Although the training and test classes share the same set of prototypes, the actual distributions of project prototypes during training and testing could be much different due to the distribution shift between the training and testing classes. During our coded residual transform, we assign different weights for different project residuals based on the correlation between the feature and the corresponding prototype. Therefore, for training classes, the subset of prototypes which are close to the training images will have larger weights. Similarly, for the testing classes, the subset of prototypes which are close to the test images will have larger weights. This correlation-based weighting for the projection residual contribute significantly to the overall performance gain.

## 2 Related Work and Unique Contributions

This work is related to deep metric learning, transformer-based learning methods, and residual encoding. In this section, we review the existing methods on these topics and discuss the unique novelty of our approach.

**(1) Deep metric learning.** Deep metric learning aims to learn discriminative features with the goal to minimize intra-class sample distance and maximize inter-class sample distance in a contrastive manner. Contrastive loss [30] has been successfully used in early methods of deep metric learning, aiming to optimize pairwise distance of between samples. By exploring more sophisticated relationship between samples, a variety of metric loss functions, such as triplet loss [31], lifted structured loss [32], proxy-anchor loss [9], and multi-similarity (MS) loss [11], have been developed. According to the studies in [22] and [33], the MS loss was verified to be one of the most efficient metric loss functions. Some recent methods explore how to use multiple features to learn robust feature embeddings. Kan *et al.* [2] and Seidenschwarz *et al.* [18] adopt K-nearest neighbors (k-NN) of an anchor image to build local graph neural network (GNN) and refine embedding vectors based on message exchanges between the graph nodes. Zhao *et al.* [17] proposed a structural matching method to learn a metric function between feature maps based on the optimal transport theory. Based on the form of softmax, margin-based softmax loss functions [34, 35, 36] were also proposed to learn discriminative features. Sohn [37] improved the contrastive loss and triplet loss by introducing $N$ negative examples and proposed the N-pair loss function to speed up model convergence during training.

**(2) Transformer-based learning methods.** This work is related to transformer-based learning methods since our method also analyze the correlation between features and uses this correlation information to aggregate features. The original work of transformer [27] aims to learn a self-attention function and a feed forward transformation network for nature language processing. Recently, it has been successfully applied to computer vision and image processing. ViT [38] demonstrates that a pure transformer can achieve state-of-the-art performance in image classification. ViT treats each image as a sequence of tokens and then feeds them to multiple transformer layers to perform the classification. Subsequently, DeiT [39] further explores a data-efficient training strategy and a distillation approach for ViT. More recent methods such as T2T ViT [29], TNT [40], CrossViT [41] and LocalViT [42] further improve the ViT method for image classification. PVT [43] incorporates a pyramid structure into the transformer for dense prediction tasks. After that, methods such as Swin [28], CvT [44], CoaT [45], LeViT [46], Twins [47] and MiT [48] enhance the local continuity of features and remove fixed size position embedding to improve the performance of transformers for dense prediction tasks. For deep metric learning, El-Nouby *et al.* [1] and Ermolov *et al.* [20] adopt the DeiT-S network[39] as a backbone to extract features, achieving impressive performance.

**(3) Residual encoding.** Residual encoding was first proposed by Jégou *et al.* [49], where the vector of locally aggregated descriptors (VLAD) algorithm is used to aggregate the residuals between features and their best-matching codewords. Based on the VLAD method, VLAD-CNN [50] has developed the residual encoders for visual recognition and understanding tasks. NetVLAD [51] and Deep-TEN [52] extend this idea and develop an end-to-end learnable residual encoder based on soft-assignment. It should be noted that features learned by these methods typically have very large sizes, for example, 16k, 32k and 4096 for AlexNet [53], VGG-16 [54] and ResNet-50 [55], respectively.

**(4) Unique Contributions.** Compared to the above existing methods, the unique contributions of this paper can be summarized as follows: (1) We introduce a new CRT method which learns a set of prototype features, project the feature map onto each prototype, and then encode its features using their projection residuals weighted by their correlation coefficients with each prototype. (2) We introduce a diversity constraint for the set of prototype features so that the CRT method can represent and encode the feature map from a set of complimentary perspectives. Unlike existing transformer-based feature representation approaches which encode the original values of features based on global correlation analysis, the proposed coded residual transform encode the relative differences between original features and their projected prototypes. (3) To further enhance the generalization performance, we propose to enforce the feature distribution consistency between coded residual transforms with different sizes of projection prototypes and embedding dimensions. (4) We demonstrate that this multi-perspective projection with diversified prototypes and coded residual representation based on relative differences are able to achieve significantly improved generalization

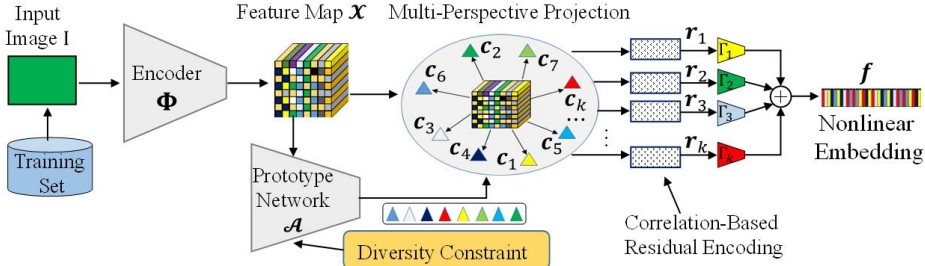

Figure 1: Overview of the proposed Coded residual transform (CRT) for generalizable deep metric learning.

capability in metric learning. Our proposed CRT method outperforms the state-of-the-art methods by large margins and improving upon the baseline method by up to 4.28%.

## 3  Method

In the following sections, we present our proposed method of coded residual transform for deep metric learning.

### 3.1  Method Overview

Figure 1 provides an overview of the proposed CRT method. We first use a backbone network $\Phi$ to encode the input image and extract its feature map $\mathcal{X} \in \mathbb{R}^{H \times W \times L}$, which consists of $H \times W$ feature vectors of size $L$. To improve the generalization capability of deep metric learning on feature map, we learn a prototype network to generate a set of diversified prototype features $\{\mathbf{c}_1, \mathbf{c}_2, \cdots, \mathbf{c}_K\}$ from the training set. Here, by "diversified" we mean these prototypes are able to provide multi-perspective complimentary representation of image features. We then project all feature vectors in the feature map onto each prototype $\mathbf{c}_k$ and encode them using the projection residuals weighted by their correlation coefficients with respect to this prototype. Each prototype $\mathbf{c}_k$ will generate a coded residual representation $\mathbf{r}_k$ of the feature map. After being further processed by a nonlinear embedding network $\boldsymbol{\Gamma}_k$, these coded residual representations will be added together to form the final nonlinear feature embedding. The above feature embedding process is referred to as *coded residual transform*. To further enhance the generalization performance, we introduce a consistency constraint between two different coded residual transform with different values of $K$ which is the size of the prototype set. In the following sections, we will explain the above major components of our CRT method in more details.

### 3.2  Multi-Perspective Projection and Correlation-Based Residual Encoding

Given an input image $I$, a backbone network $\Phi$ is used to encode image $I$ into a feature map $\mathcal{X} \in \mathbb{R}^{H \times W \times L}$, i.e., $\mathcal{X} = \Phi(I)$. First, we learn prototype network to generate a set of diversified prototype features $\mathcal{C} = \{\boldsymbol{c}_1, \boldsymbol{c}_2, \cdots, \boldsymbol{c}_K\}, \boldsymbol{c}_k \in \mathbb{R}^L$, whose learning process will be explained in the following section. For each prototype $\boldsymbol{c}_k$, we project the whole feature map $\mathcal{X}$ onto this prototype. Specifically, for each feature $\boldsymbol{x}_j \in \mathcal{X}$ inside the feature map, we find the relative difference $\boldsymbol{d}_{kj} = \boldsymbol{x}_j - \boldsymbol{c}_k$ between the feature $\boldsymbol{x}_j$ and the prototype $\boldsymbol{c}_k$. Let $w_{kj} = \boldsymbol{x}_j \cdot \boldsymbol{c}_k^T$ be their correlation coefficient. Then, with prototype $\boldsymbol{c}_k$, the whole map is transformed into the following feature vector using weighted summation of $\boldsymbol{d}_{kj}$:

$$\boldsymbol{r}_k = \sum_{j=1}^{H \times W} \log[1 + \exp(\boldsymbol{x}_j^T \boldsymbol{c}_k)](\boldsymbol{x}_j - \boldsymbol{c}_k), \tag{1}$$

where the nonlinear function $\log[1 + \exp(w_{kj})]$ converts the correlation coefficient $w_{kj}$ into a positive number. In this way, each prototype $\boldsymbol{c}_k$ will generate a separate coded residual representation $\boldsymbol{r}_k$ of the feature map for the input image. Finally, $\boldsymbol{r}_k$ is further transformed by a nonlinear embedding

network $\mathbf{\Gamma}_k$ with multiple fully connected layers and one GELU nonlinear activation layer [56]. By aggregating those nonlinear embeddings, the final embedded feature $\boldsymbol{f}$ is then given by

$$\boldsymbol{f} = \frac{1}{K}\sum_{k=1}^{K}\mathbf{\Gamma}_k(\mathbf{r}_k). \tag{2}$$

The task of the prototype network is to generate an optimized set of prototypes. Its input is the features from the feature map. Its weights are the prototypes that need to be learned, and its output is the correlations between the input features and the weights, which are the correlations between the feature map and prototype. The prototype network will update these prototypes using stochastic gradient descent (SGD) algorithm in back propagation. In this work, we use an one-layer fully connected network to implement this prototype network. The regularization of prototypes will be discussed in the following section.

Figure 2 shows two sets of examples of correlation map between different prototypes and the feature map. Each image corresponds to one prototype. The heat map shows the correlation between the prototype and the feature at the corresponding spatial location from the feature map. We can see that different prototypes are capturing different semantic components of the image and the feature map is projected and encoded from a large set of different semantic perspectives. Our generalization analysis and ablation studies will demonstrate that this multi-perspective projection and encoding will provide significantly enhanced generalization capability and improved metric learning performance.

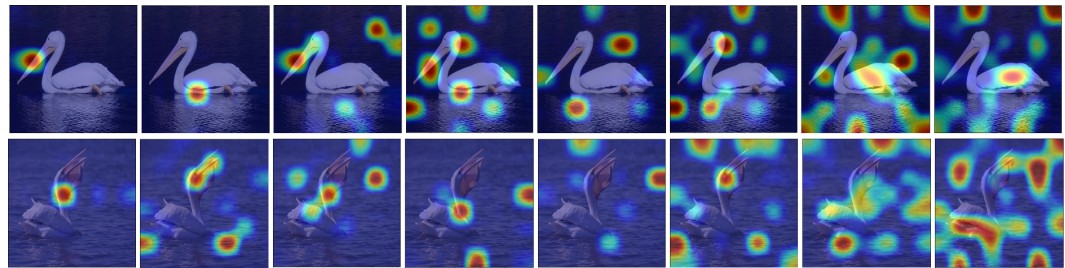

Figure 2: Correlation heat maps between a learned set of prototypes (corresponding to 8 prototypes) and feature maps of two images. We can see that different prototypes response to different local or global views. The red dots on the top right corner indicates the background prototype.

### 3.3 Network Model Learning and Generalization Analysis

As illustrated in Figure 1, our proposed CRT method involves the encoder network $\mathbf{\Phi}$, the prototype network $\mathcal{A}$, and the multi-perspective feature embedding networks $\{\mathbf{\Gamma}_k\}$. These networks are jointly trained in an end-to-end manner. The task of the prototype network $\mathcal{A}$ is to generate a set of prototypes for efficient projection of the input features. From our experiments, we find that it is important to enforce diversity among these prototypes so that they can provide a diversified and robust representation of the image features. Diversity means that the prototype features have small correlation with each other. Let $\mathbf{C} = \{\boldsymbol{c}_1, \boldsymbol{c}_2, \cdots, \boldsymbol{c}_K\}$ be a set of prototypes generated by the prototype network. When training the prototype network, we need to minimize the following average correlation between these prototype features, which is defined to be the prototype diversity loss

$$L_{DIV} = \frac{1}{K(K-1)}\sum_{k\neq j}\frac{|\boldsymbol{c}_k \cdot \boldsymbol{c}_j^T|}{\|\boldsymbol{c}_k\|_2 \cdot \|\boldsymbol{c}_j\|_2}, \tag{3}$$

where the cosine similarity is used to measure the correlation between two different prototype features. To train the overall feature embedding network, we use the multi-similarity (MS) loss [11], denoted by $\mathrm{L}_{MS}$, as our metric loss function to minimize the similarity for negative pairs and maximize the similarity for positive pairs. Here, negative pairs are samples from different classes while positive pairs are samples from the same class. The prototype diversity loss $L_{DIV}$ and the MS loss $\mathrm{L}_{MS}$ are combined to form the overall loss function $L = L_{DIV} + \lambda_1\mathrm{L}_{MS}$ for end-to-end training of our CRT network.

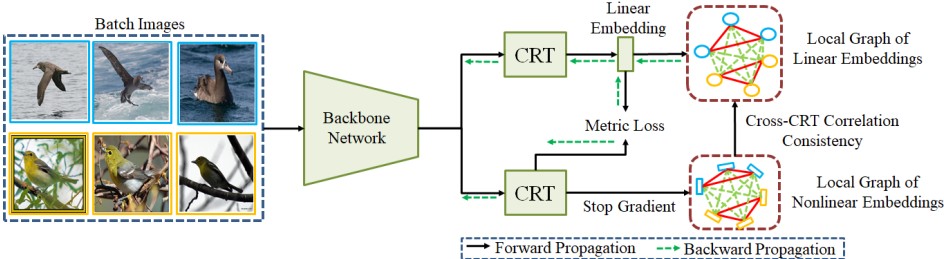

Figure 3: The training method of the proposed framework.

Table 1: The embedding space density ($\uparrow$) on the experimental datasets.

| Method | CUB | Cars | SOP | In-Shop |
|---|---|---|---|---|
| Baseline | 0.72 | 0.79 | 0.38 | 0.28 |
| +CRT | **0.90** | **1.01** | **0.40** | **0.33** |

Table 2: The spectral decay ($\downarrow$) on the experimental datasets.

| Method | CUB | Cars | SOP | In-Shop |
|---|---|---|---|---|
| Baseline | 0.27 | 0.24 | 0.31 | 0.23 |
| +CRT | **0.19** | **0.15** | **0.13** | **0.10** |

**Cross CRT consistency loss.** To further enhance the generalization performance, we propose to enforce the feature distribution consistency between coded residual transforms with different sizes of projection prototypes. As shown in Figure 3, we construct two CRT embedding branches with different number of projection prototypes. For example, in our experiment, the first branch has an prototype set size $K_1 = 49$ with an embedded feature size of 128, while the second branch has an prototype set size $K_2 = 64$ with an embedded feature size of 1024. Suppose that the current training batch of images are $\mathcal{I} = \{I_1, I_2, \cdots, I_N\}$. Let the feature embeddings of the first CRT branch be $\{\boldsymbol{f}_1^{(1)}, \boldsymbol{f}_2^{(1)}, \cdots, \boldsymbol{f}_N^{(1)}\}$ and the feature embeddings of the second branch be $\{\boldsymbol{f}_1^{(2)}, \boldsymbol{f}_2^{(2)}, \cdots, \boldsymbol{f}_N^{(2)}\}$.

Define the similarity matrix

$$\mathbb{S}^{(1)} = \left[\mathbf{s}_{ij}^{(1)}\right]_{1 \le i,j \le N}, \quad \mathbf{s}_{ij}^{(1)} = \frac{\boldsymbol{f}_i^{(1)} \cdot [\boldsymbol{f}_j^{(1)}]^T}{\|\boldsymbol{f}_i^{(1)}\|_2 \cdot \|\boldsymbol{f}_j^{(1)}\|_2}. \tag{4}$$

Similarly, we can define the similarity matrix for the second branch as $\mathbb{S}^{(2)}$. The consistency between these two CRT branches are defined to be the $L_1$ distance between these two similarity matrices

$$L_{CON} = \|\mathbb{S}^{(1)} - \mathbb{S}^{(2)}\|_1. \tag{5}$$

This consistency loss is then combined with the original loss function for the CRT network. Thus, the overall loss is given by $L = L_{DIV} + \lambda_1 \mathrm{L}_{MS} + \lambda_2 \cdot L_{CON}$.

**Generalization capability analysis.** In this section, we analyze the generalization capability of our proposed CRT method. Roth *et al.* [22] established two metrics to measure the generalization capability of deep metric learning methods: (1) embedding space density and (2) spectral decay. Methods with higher embedding space density and smaller spectral decay often have better generalization capabilities. The embedding space density metric $\mathcal{D}_{\text{ESD}}$ is defined as the ratio between the average of intra-class distance $\mathcal{D}_{\text{Intra}}$ and the average of inter-class distance $\mathcal{D}_{\text{Inter}}$:

$$\mathcal{D}_{\text{ESD}} = \mathcal{D}_{\text{Intra}}/\mathcal{D}_{\text{Inter}}. \tag{6}$$

The spectral decay metric $\rho_{\text{SD}}$ is defined to be the KL-divergence between the spectrum of $d$ singular values $V^{\text{SV}}$ (obtained from Singular Value Decomposition, SVD) and a $d$-dimensional uniform distribution $\mu_d$. It is inversely related to the entropy of the embedding space:

$$\rho_{\text{SD}} = \mathcal{D}_{\text{KL}}(\mu_d, V^{\text{SV}}). \tag{7}$$

Table 1 and Table 2 show the embedding space density and the spectral decay values of the feature embedding learned by the baseline method and our CRT method on the test datasets used in our experiments. We can see that our method achieves significantly higher embedding space density and much lower spectral decay when the model tested on unseen classes, which indicates significantly improved generalization capability.

# 4 Experimental Results

Following the same procedure of existing papers [2, 7, 13, 18, 20], we evaluate the performance of our proposed CRT method on four benchmark datasets for image retrieval tasks.

## 4.1 Datasets and Experimental Settings

**(1) Datasets.** In the following experiments, we use the same benchmark datasets as in existing papers for direct performance comparison. Four datasets, i.e., CUB-200-2011 [57], Cars-196 [58], Standford Online Products (SOP) [32], and In-Shop Clothes Retrieval (In-Shop) [59], are used to our experiments. We use the same training and test split as in existing papers. Hyperameters were determined previous to the result runs using a 80-20 training and validation split.

**(2) Performance metrics and experimental settings.** Following the standard protocol in [2, 7, 13, 18, 20, 60], the Recall@K [61] is used to evaluate the performance of our algorithm. For all datasets, our method is evaluated with the image data only, without using the bounding box information. During training, we randomly sample a set of images in each iteration to train the network. In each iteration, we first randomly choose the image classes, and then randomly sample 5 images from each class to form a batch. For the CUB and Cars datasets, we sampled 16 classes in a batch. For the SOP and In-Shop datasets, we sampled 36 classes in a batch. We apply random cropping with random flipping and resizing to $227 \times 227$ for all training images. For testing, we only use the center-cropped image to compute the feature embedding. Unless otherwise specified, we use the MixTransformer-B2 (MiT-B2) developed in [48] as our baseline encoder network with its pyramid MLP head being removed. Our CRT algorithm operates on the feature map generated by the last block (the 4-th block) of the backbone encoder network. The initial learning rate is $3e^{-5}$. For all images, the MS loss weight $\lambda_1$ in the first embedding branch is set as 1.0. In second embedding branch, it is set as 0.1 for the CUB and Cars datasets, and 0.9 for the SOP and In-Shop datasets. The consistency loss weight $\lambda_2$ is set to 0.9. The backbone network is pre-trained on the ImageNet-1K dataset.

## 4.2 Performance Comparisons with the State-of-the-art Methods

We compare the performance of our CRT method with the state-of-the-art deep metric learning methods. Performance comparison results on the CUB and Cars datasets are shown in Table 3. Table 4 shows the comparison results on the SOP and In-Shop datasets. It should be noted that the feature dimension of our CRT method is set to be 128. We can see that our CRT method significantly outperforms the current best deep metric learning method Hyp-DeiT [20]. The results of the Hyp-DeiT method are cited under the same experimental settings where the embedding size is 128 with image size of $227 \times 227$, and a pretrained model on the ImageNet-1K. We choose this setting so that we can compare our method with many recent papers on metric learning. For example, on the CUB dataset, the top-1 recall rate has been improved by 4.28%. The experimental results on the rest datasets suggest that our CRT method consistently outperforms other deep metric learning methods by large margins.

As analyzed in [22] and [33], most of the deep metric learning methods use different experimental conditions, such as different settings of batch size, data augmentation methods, embedding sizes, training strategies, backbone networks, and input image sizes, to obtain the state-of-the-art performance, which could lead to confusion on the effectiveness of those methods. For example, Hyp-DeiT [20] and IRT$_R$ [1] used larger batch sizes and memory bank technique to obtain better performance. The method of XBM+RTT [21] used the re-rank technique to obtain the highest top-1 recall rate on the SOP dataset. The Group Loss++ [7] also used the re-rank method to improve the retrieval performance. Moreover, Proxy-Anchor [9], Group Loss++ [7] and ETLR [13] used mixed pooling methods to improve embedding performance. During our experiments, we find that it is challenging to re-produce their original results with our available computing resources and to conduct fair comparisons, especially for those methods without publicly available source code. Instead, we implement our CRT method with the MS method [11] which has been extensively used in existing deep metric learning methods. In the following, we provide a fair comparison between our method and the baseline MS method with different backbone networks under the same experimental conditions.

Table 3: Comparison of retrieval performance on the CUB and Cars datasets.

| Methods | Dim | CUB | | | | Cars | | | |
|---|---|---|---|---|---|---|---|---|---|
| | | R@1 | R@2 | R@4 | R@8 | R@1 | R@2 | R@4 | R@8 |
| A-BIER [TPAMI20] [60] | 512 | 57.5 | 68.7 | 78.3 | 82.6 | 82.0 | 89.0 | 93.2 | 96.1 |
| MS [CVPR19] [11] | 512 | 65.7 | 77.0 | 86.3 | 91.2 | 84.1 | 90.4 | 94.0 | 96.5 |
| Proxy-Anchor [CVPR20] [9] | 512 | 68.4 | 79.2 | 86.8 | 91.6 | 86.1 | 91.7 | 95.0 | 97.3 |
| DRML-PA [ICCV21] [62] | 512 | 68.7 | 78.6 | 86.3 | 91.6 | 86.9 | 92.1 | 95.2 | 97.4 |
| ETLR [CVPR21] [13] | 512 | 72.1 | 81.3 | 87.6 | - | 89.6 | 94.0 | 96.5 | - |
| DCML-MDW [CVPR21] [63] | 512 | 68.4 | 77.9 | 86.1 | 91.7 | 85.2 | 91.8 | 96.0 | 98.0 |
| D & C [TPAMI21] [64] | 512 | 68.4 | 78.7 | 86.0 | 91.6 | 87.8 | 92.5 | 95.4 | - |
| IBC [ICML21] [18] | 512 | 70.3 | 80.3 | 87.6 | 92.7 | 88.1 | 93.3 | 96.2 | 98.2 |
| LSCM-GNN [TIP22] [2] | 512 | 68.5 | 77.3 | 85.3 | 91.3 | 87.4 | 91.5 | 94.9 | 97.0 |
| Group Loss++ [TPAMI22] [7] | 512 | 72.6 | 80.5 | 86.2 | 91.2 | 90.4 | 93.8 | 96.0 | 97.5 |
| IRT$_R$ [arXiv21] [1] | 384 | 74.7 | 82.9 | 89.3 | 93.3 | - | - | - | |
| PA+DIML [ICCV21] [17] | 128 | 66.46 | - | - | - | 86.13 | - | - | - |
| Hyp-DeiT[CVPR22] [20] | 128 | 74.7 | 84.5 | 90.1 | 94.1 | 82.1 | 89.1 | 93.4 | 96.3 |
| **Ours**: CRT | 128 | **78.98** | **86.68** | **91.61** | **95.04** | **91.16** | **94.92** | **96.79** | 98.03 |
| **Ours**: Gain | 128 | 4.28 | 2.18 | 1.51 | 0.94 | 0.76 | 0.92 | 0.29 | −0.17 |

Table 4: Comparison of retrieval performance on the SOP and In-Shop datasets.

| Methods | Dim | SOP | | | | In-Shop | | | |
|---|---|---|---|---|---|---|---|---|---|
| | | R@1 | R@10 | R@100 | R@1000 | R@1 | R@10 | R@20 | R@30 |
| Fusing-Net [TIP19] [8] | 512 | 71.8 | 86.3 | 94.1 | 98.2 | 82.4 | 95.1 | 96.7 | 97.4 |
| A-BIER [TPAMI20] [60] | 512 | 74.2 | 86.9 | 94.0 | 97.8 | 83.1 | 95.1 | 96.9 | 97.5 |
| MS [CVPR19] [11] | 512 | 78.2 | 90.5 | 96.0 | 98.7 | 89.7 | 97.9 | 98.5 | 98.8 |
| DRML-PA [ICCV21] [62] | 512 | 71.5 | 85.2 | 93.0 | - | - | - | - | - |
| ETLR [CVPR21] [13] | 512 | 79.8 | 91.1 | 96.3 | - | - | - | - | - |
| DCML-MDW [CVPR21] [63] | 512 | 79.8 | 90.8 | 95.8 | - | - | - | - | - |
| D & C [TPAMI21] [64] | 512 | 79.8 | 90.4 | 95.2 | - | 90.4 | 97.6 | - | - |
| IBC [ICML21] [18] | 512 | 81.4 | 91.3 | 95.9 | - | 92.8 | 98.5 | 99.1 | 99.2 |
| LSCM-GNN [TIP22] [2] | 512 | 79.7 | 90.5 | 95.7 | 98.4 | 92.4 | 98.5 | 99.1 | 99.3 |
| Group Loss++ [TPAMI22] [7] | 512 | 79.2 | 90.1 | 95.8 | - | 90.9 | 97.6 | 98.4 | 98.9 |
| IRT$_R$ [arXiv21] [1] | 384 | 84.0 | 93.6 | 97.2 | 99.1 | 91.5 | 98.1 | 98.7 | 99.0 |
| PA+DIML [ICCV21] [17] | 128 | 79.22 | - | - | - | - | - | - | - |
| XBM+RTT [ICCV21] [21] | 128 | 84.5 | 93.2 | 96.6 | 99.0 | - | - | - | - |
| Hyp-DeiT[CVPR22] [20] | 128 | 83.0 | 93.4 | 97.5 | 99.2 | 90.9 | 97.9 | 98.6 | 98.9 |
| **Ours**: CRT | 128 | 83.41 | **93.86** | **97.66** | **99.31** | **94.48** | **99.37** | **99.68** | **99.75** |
| **Ours**: Gain | 128 | −1.09 | 0.26 | 0.16 | 0.11 | 1.68 | 0.87 | 0.58 | 0.45 |

## 4.3 Performance Comparisons with Different Backbone Networks.

We conduct experiments with different backbone networks to verify the effectiveness of the proposed CRT method. For fair comparisons [22, 33], we reproduce the MS method [11] and compare the performance using the same experimental settings. The hyper-parameter settings are consistent with the original paper [11]. The batch size is set to 80 on the CUB and Cars datasets, and 180 on the SOP and In-Shop datasets. The embedded feature size is set to 128. Results are shown in Table 5, we can see that our proposed method is able to consistently improve the performance of the baseline method. It should be noted that the DeiT-S and MiT-B1 networks are both based on the transformer, a powerful method for exploiting the global correlation between features for very efficient image representation. From Table 5, we can see that, even on top of these two baseline methods with the powerful transformer backbone network, our coded residual transform is still able to improve the metric learning performance by up to 3.7%, which is quite impressive. This is because our CRT method introduces the new ideas of multi-perspective projection and residual encoding, which have significantly improved the generalization performance of the deep metric learning.

## 4.4 Ablation Studies

In this section, we provide extensive ablation studies to further understand our proposed CRT method and characterize the performance of major algorithm components. In the following experiments, unless otherwise specified, the batch size $N$ is set to 80. The backbone network used in this experiment is the MiT-B1.

Table 5: Comparisons of Recall@K (%) on the CUB, Cars, SOP and In-Shop datasets for different backbone networks.

| Backbones | Methods | CUB | | | Cars | | | SOP | | | In-Shop | | |
|---|---|---|---|---|---|---|---|---|---|---|---|---|---|
| | | R@1 | R@2 | R@4 | R@1 | R@2 | R@4 | R@1 | R@10 | R@100 | R@1 | R@10 | R@20 |
| GoogLeNet | MS | 56.53 | 69.13 | 79.54 | 76.49 | 84.76 | 90.38 | 70.10 | 85.73 | 94.41 | 87.33 | 97.73 | 98.56 |
| | +CRT | **59.23** | **71.02** | **81.52** | **78.50** | **86.35** | **91.91** | **71.06** | **86.41** | **94.73** | **88.31** | **97.83** | **98.68** |
| | Gain | **2.7** | **1.89** | **1.98** | **2.01** | **1.59** | **1.53** | **0.96** | **0.68** | **0.32** | **0.98** | **0.1** | **0.12** |
| BN-Inception | MS | 61.95 | 72.59 | 82.44 | 80.59 | 87.50 | 92.63 | 74.10 | 88.46 | 95.43 | 90.75 | 98.52 | 99.06 |
| | +CRT | **65.78** | **76.72** | **85.31** | **81.38** | **88.38** | **93.08** | **75.65** | **89.04** | **95.64** | **91.52** | **98.61** | **99.19** |
| | Gain | **3.83** | **4.13** | **2.87** | **0.79** | **0.88** | **0.45** | **1.55** | **0.58** | **0.21** | **0.77** | **0.09** | **0.13** |
| ResNet-50 | MS | 62.64 | 73.73 | 83.20 | 79.92 | 87.63 | 92.41 | 76.49 | 89.33 | 95.66 | 90.11 | 97.52 | 98.27 |
| | +CRT | **64.20** | **75.54** | **84.12** | **83.29** | **89.76** | **93.88** | **78.97** | **91.10** | **96.51** | **92.38** | **98.77** | **99.25** |
| | Gain | **1.56** | **1.89** | **0.92** | **3.37** | **2.13** | **1.47** | **2.48** | **1.77** | **0.85** | **2.27** | **1.25** | **0.98** |
| DeiT-S | MS | 72.32 | 82.43 | 89.20 | 82.20 | 89.34 | 93.76 | 79.56 | 91.63 | 96.87 | 92.47 | 98.72 | 99.21 |
| | +CRT | **74.71** | **83.83** | **89.65** | **84.26** | **90.95** | **94.90** | **81.60** | **92.65** | **97.20** | **93.31** | **98.98** | **99.35** |
| | Gain | **2.39** | **1.4** | **0.45** | **2.06** | **1.61** | **1.14** | **2.04** | **1.02** | **0.33** | **0.84** | **0.26** | **0.14** |
| MiT-B1 | MS | 72.25 | 81.85 | 88.54 | 87.36 | 92.26 | 95.23 | 79.67 | 91.55 | 96.66 | 92.20 | 98.64 | 99.21 |
| | +CRT | **75.95** | **84.47** | **90.26** | **89.60** | **94.20** | **96.47** | **82.32** | **93.02** | **97.19** | **93.51** | **99.11** | **99.50** |
| | Gain | **3.7** | **2.62** | **1.72** | **2.24** | **1.94** | **1.24** | **2.65** | **1.47** | **0.53** | **1.31** | **0.47** | **0.29** |

Table 6: The Recall@K (%) for different component on the CUB dataset.

| Model | CUB | | | |
|---|---|---|---|---|
| | R@1 | R@2 | R@4 | R@8 |
| The Proposed Method | **75.95** | **84.47** | 90.26 | **94.51** |
| −CRT | 73.97 | 84.03 | **90.41** | 94.31 |
| −CRT-Consistency | 72.25 | 81.85 | 88.54 | 93.57 |

Table 7: The Recall@K (%) with (w) and without (w/o) using multi-perspective CRT feature transformation.

| | CUB | | | |
|---|---|---|---|---|
| | R@1 | R@2 | R@4 | R@8 |
| w/o | 74.34 | 83.59 | 90.23 | 94.14 |
| w | **75.95** | **84.47** | **90.26** | **94.51** |

**(1) Contributions of major algorithm components.** From the performance evaluation perspective, our algorithm has two major components, the coded residual transform (CRT) and cross-CRT correlation consistency (CRT-Consistency). We fix the MS loss weight $\lambda_1$ in the first and second embedding branches as 1.0 and 0.1, respectively. The consistency loss weight $\lambda_2$ is set to be 0.9. From Table 6, we can see that the performance dropped 1.98% for the top-1 recall rate without the CRT. The performance future dropped 1.69% for the top-1 recall rate without the CRT-Consistency, which is the baseline result that obtained based on the MS loss function. Moreover, the multi-perspective CRT features transformation plays an important role for the final performance. For different projection prototypes, the CRT features can be transformed using different embedding networks or the same embedding network. Here, we compare the performance of our CRT method with and without using the same embedding network for the transformation of CRT features. Results are shown in Table 7. We can see that better performance can be obtained using different embedding networks for each CRT feature corresponding to each projection prototype. This shows the effectiveness of our multi-perspective projection.

**(2) Impact of the projection prototypes.** In this experiment, we analyze the impact of different numbers of projection prototypes. Table 8 and Table 9 show the results of different numbers of projection prototypes for the first embedding branch (low-dimensional embedding branch) and the second embedding branch (high-dimensional embedding branch), respectively. In Table 8, the number of projection prototypes in the second embedding branch is fixed at 64. This table reports the results of embedding computed from the first embedding branch when the number of projection prototypes changed from 1 to 64. The weights of the backbone networks are shared and the weights of the CRT heads are not shared for this experiment. In Table 9, the number of project prototypes are equal for the first and the second CRT embedding branches, which are changed from 1 to 100, and the weights of both the backbone network and the CRT branches are shared in this experiment. We can see that when the number of prototypes is set to be 64, the embedding performance reaches to the best.

In Table 9, the network weight of the second embedding branch is shared from the first embedding branch. Here, we compare the performance of our CRT method with and without shared weights between these two embedding branches. Results are shown in Table 10. We can see that the performance remains almost the same between these two study cases. In other experiments, to decrease the memory consumption, we shared weights for all the experiments.

Table 8: The Recall@K (%) for different number of projection prototypes in the first embedding branch.

| $K_1$ | CUB | | | |
|---|---|---|---|---|
| | R@1 | R@2 | R@4 | R@8 |
| 1 | 75.76 | 84.52 | 90.72 | **94.78** |
| 4 | 75.57 | 84.30 | 90.56 | 94.33 |
| 16 | 75.44 | 84.25 | 90.24 | 94.46 |
| 64 | **75.96** | **84.60** | **90.75** | 94.31 |

Table 9: The Recall@K (%) for different number of projection prototypes in the second embedding branch.

| $K_2$ | CUB | | | |
|---|---|---|---|---|
| | R@1 | R@2 | R@4 | R@8 |
| 1 | 74.26 | 84.30 | 89.82 | 93.96 |
| 4 | 74.54 | 84.03 | 90.06 | 94.01 |
| 16 | 74.63 | 83.74 | 90.24 | 94.02 |
| 64 | **75.95** | **84.47** | 90.26 | **94.51** |
| 100 | 75.15 | 84.35 | **90.36** | 94.31 |

Table 10: The Recall@K (%) with (w) and without (w/o) shared weights between these two embedding branches.

| | CUB | | | |
|---|---|---|---|---|
| | R@1 | R@2 | R@4 | R@8 |
| w | 75.95 | 84.47 | 90.26 | **94.51** |
| w/o | **75.96** | **84.60** | **90.75** | 94.31 |

## 5 Conclusion

In this paper, we have developed a coded residual transform for generalizable deep metric learning, which consists of a multi-perspective projection and coded residual transform encoder and a cross-CRT correlation consistency constraint. It has two unique characteristics. First, it represents and encodes the feature map from a set of complimentary perspectives based on projections onto diversified prototypes. Second, unlike existing transformer-based feature representation approaches which encode the original values of features based on global correlation analysis, the proposed coded residual transform encodes the relative differences between original features and their projected prototypes. The proposed CRT method has achieved new state-of-the-art metric learning performance on benchmark datasets. We hope our method can motivate further research. One limitation is that the memory and compute usage will be increased during training for these two embedding branches, and we shared weights between them to solve this problem in our experiments. Another limitation is that the projection prototypes were learned from the training set. It is unclear whether it is the best projection prototypes for new test classes. We leave it for the future work.

## Acknowledgments

This work was supported in part by the National Key R&D Program of China 2021YFE0110500, in part by the National Natural Science Foundation of China under Grant 62202499, 61872034, 62062021 and 62011530042, in part by the Hunan Provincial Natural Science Foundation of China under Grant 2022JJ40632, in part by the Beijing Municipal Natural Science Foundation under Grant 4202055.

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
