# OpenReview forum: "Coded Residual Transform for Generalizable Deep Metric Learning"
_NeurIPS.cc/2022/Conference — NeurIPS 2022 Accept_

### Official Review · Reviewer_HDjt · 2022-06-27

**Rating:** 7
**Confidence:** 5
**Soundness:** 3 good
**Presentation:** 3 good
**Contribution:** 3 good

**Summary:**

This paper revives a classic image retrieval technique, VLAD, with modern Deep Network and show promising results. In short for the main idea of the paper, the proposed code residual transform (CRT) method maps the feature map into anchor pools and output residue features to represent an input image. In table 5, CRT outperforms the baseline method by a clear gap across multiple datasets. Instead only providing the number to beat the SOTA, the author also provide additional evidence such as feature visualization and embedding space density metric to support the proposed method.

**Questions:**

1. In the figure3 visualization, can you explain the red dots on the top right corner of image in the end of first row?
2. Would you like provide the correlation/similarity matrix for the anchor pool? It is supposed to be independence as much as possible as show in Eq 3. But as shown in figure 3, different attentions are kind of overlap.
3. can you provide the sampler detail? what is the batch size? how many classes in a batch?


**Ethics Review Area:**

["I don’t know"]

**Strengths And Weaknesses:**

Strengths
Originality: The proposed method revives a classic image retrieval technique, VLAD, with modern Deep Network is novel
Quality: The submission is technically sound. The anchor pool is well visualized in Figure 3(But still some flaw, see later). Additional metric such as table 1 embedding space density which helps understanding where is gain coming from with the proposed method
Clarity: The submission is clearly written and well organized.
Significance: Reviving a classic image retrieval technique with modern techniques is significant.

Weaknesses
Clarity: Figure 1 and Figure 2 should be merged together. Figure 1 is clear enough for presenting the idea, but figure 2 is not.
Clarity: Please use another terminology for the "anchor" in the main paper. The term anchor is already used in the triplet loss. It will confuse readers who are familiar with DML.

---

> ### Author Response · Authors · 2022-08-02
> **Response to the comment on Figures 1 and 2, the terminology of anchor, Figure 3, the similarity matrix for the anchor pool and sampler detail.**
>
> We really appreciate your thorough and insightful review of our paper! We also thank your positive and encouraging comments about our paper. In the following, we provide detailed response to your comments.
>
> $\textbf{1. Response to the comment on Figures 1 and 2: }$
>
> Thanks for this comment! In the revised paper, we have followed your comment and moved Figure 2 to the supplemental materials in the revised paper.
>
> $\textbf{2. Response to the comment on the terminology of anchor:}$
>
> Thanks for this valuable comment! At the end of this discussion with more feedback from you, we will follow your comment to replace the anchor with another word, such as prototype.
>
> $\textbf{3. Response to comment on Figure 3 (now Figure 2):}$
>
> Thanks for pointing out this! In Figure 2 (Figure 3 in the original paper), the red dots on the top right corner indicates the background anchor. These examples show the diversity of the learned anchors. We have emphasized this in the revised paper (Figure 2).
>
> $\textbf{4. Response to the comment on correlation/similarity matrix for the anchor pool:}$
>
> Thanks for this helpful comment! In the revised paper, following your comment, we have added a new figure, Figure 11 in the Supplemental Materials to visualize the correlation matrix. We can see that most of the similarities between anchors are very low, only few similarities computed between anchors are relatively large. This result shows that most of the anchors learned by our method are independent.
>
> $\textbf{5. Response to the comment on the sampler detail, batch size, and classes in a batch:}$
>
> Thanks for this comment! We have added the description of the batch samplers for training in the revised paper (Section 4.1). During training, we randomly sample a set of images in each iteration to train the network. In each iteration, we first randomly choose the image classes, and them randomly sample 5 images from each class to form a batch. For the CUB and Cars datasets, we sampled 16 classes in a batch. For the SOP and In-Shop datasets, we sampled 36 classes in a batch. Thus, the batch size is 80 for the CUB and Cars  datasets, and 180 for the SOP and In-Shop datasets.

---

> > ### Comment · Reviewer_HDjt · 2022-08-08
> > **Great job~**
> >
> > After reviewing the author feedback, I maintain the score to accept the paper

---

> > > ### Author Response · Authors · 2022-08-08
> > > **Thank you very much!**
> > >
> > > Thanks for your valuable comment and feedback to improve the paper, as well as your strong recommendation of our paper!
> > > - Authors

---

### Official Review · Reviewer_nhSD · 2022-06-30

**Rating:** 8
**Confidence:** 3
**Soundness:** 4 excellent
**Presentation:** 3 good
**Contribution:** 4 excellent

**Summary:**

The manuscript proposes a new framework for generalisable deep metric learning. The main idea is to learn a collection of anchors and get the final representation by transforming the original embedding vectors into their distances to the anchors, weighted by their correlation with said anchors. These distances are then aggregated by anchor-wise projection and averaging. The model is trained with Anchor Diversity Loss (minimizes anchors’ cross-correlations) and Multi-Similarity loss, which is a recent metric learning loss [11].


**Questions:**

Does the backbone get trained as well or are the weights frozen?
Was the choice of dimensionality made out of compute restrictions? I ask since other papers have used higher dimensionalities such as 384 (for ViT based models) or 512.
Related to the previous question, how did you change the output dimensionality for the ViT based methods such as Hyp-DeiT? Did you train those from scratch with an added bottleneck layer?
It is not clear to me if the second embedding branch also uses a second backbone. Does it use another backbone and is it trained separately or do they share backbones?

**Limitations:**

Since this is an image retrieval paper that makes use of common benchmarks, I do not believe there is a negative social impact to it beyond the environmental cost of training deep neural networks.

The authors only present one limitation in their conclusion: that the anchors are learned from the training set and therefore may not be the best possible anchors. I believe another limitation to be the need for two embedding branches, since in the non-sharing configuration it incurs in increased memory and compute usage (this is a lesser issue if the backbone is not duplicated)

**Strengths And Weaknesses:**

The proposed framework is sound, intuitive and the paper is easy to follow. The presented performance results are also very promising, as is the empirical support to the authors’ claims of the generalisation capability of the model (Lines 206-219). I also believe similar or modified versions of this framework could be used in the future, which makes this a good contribution.

I’ve found Table 4 to be confusing to parse given that the numbers presented here do not match with their published versions (I’ve only checked Hyp-DeIT); the authors acknowledged the discrepancy and attributed it to different settings and results that are hard to reproduce under compute constraints, which I’ve found to be fair compromises given that they have also designed the experiments in Table 5, where comparison is made fair by comparing against baseline versions of a large set of backbones.

The ablation study is complete enough and it is good to know that performance is not too sensitive to weight sharing and the number of anchors in the first embedding branch. I’ve just found it hard to parse the permutations described in Lines 297-308 and presented in tables 8-9; I don’t know what’s the number of anchors in the first branch for Table 9, or why the weights are shared for this experiment (are they not shared for Table 8?) and it raises the question of whether the weights are shared or not for the other experiments as well since the performance is comparable and the gain in compute use is likely considerable.

---

> ### Author Response · Authors · 2022-08-02
> **Response to the comment on Hyp-DeIT results, Tables 8 and 9, backbone networks and  limitations.**
>
> Thank you very much for this detailed and insightful review of our paper! We really appreciate your positive comments and recommendation of our paper! In the following, we provide detailed response to your comments.
>
> $\textbf{1. Response to the comment on Hyp-DeIT results in Table 4:}$
>
> Thanks for your comment! We cite the results from the Hyp-DeiT method under the same experimental settings (Table 2 in their original paper) where the embedding size is 128 with image size of 227$\times$227, and a pretrained model on the ImageNet-1K.  We choose this setting so that we can compare our method with many recent papers on metric learning. In the original paper of the Hyp-DeiT, improved results of Hyp-DINO and Hyp-ViT§ were reported based on a different pre-trained model, which is much different from our experimental conditions. Thus, we only include the results of Hyp-DeiT with the embedding size of 128 in our paper. In the revised paper, we have emphasized this in Section 4.2.
>
> $\textbf{2. Response to the comment on Tables 8 and 9:}$
>
> Thanks for pointing out this! Sorry that we did not emphasize this. In Table 8, the number of projection anchors in the second embedding branch is fixed at 64. This table reports the results of embedding computed from the first embedding branch when the number of projection anchors changed from 1 to 64. The weights of the backbone networks are shared and the weights of the CRT heads are not shared for this experiment. In Table 9, the number of project anchors are equal for the first and the second CRT embedding branches, which are changed from 1 to 100, and the weights of both the backbone network and the CRT branches are shared in this experiment.
>
> We can see that when the number of anchors is set to be 64, the embedding performance reaches to the best. And the performances are approximately the same for models trained with and without shared weights. In other experiments, to decrease the memory consumption, we shared weights for all the experiments.
>
> In the revised paper, we have further clarified this in Section 4.4.
>
> $\textbf{3. Response to the comment on backbone networks:}$
>
> The two embedding CRT branches are shared the same backbone network. The backbone network is pretrained on the ImageNet-1K dataset and fine-tuned during training. The embedding dimensions of the ViT based methods are changed by adding a linear layer. Here, we set the embedding size to be 128 is because recent methods are developed based on a dimension of 128 and it is important for large-scale datasets and real-time applications.
>
> In Table 5 of supplemental material, we have provided ablation studies to evaluate the impact of different embedding sizes.
>
> $\textbf{4. Response to the comment on limitations and non-sharing configurations:}$
>
> Thanks for this valuable comment! The backbone network is shared for these two embedding branches. Following your comment, in the revised paper, we have further emphasized this in Sections 4.4 and 5. For non-sharing configuration, only the weights of CRT branches are not shared, the weights of the backbone network are shared for all the experiments.

---

> > ### Comment · Reviewer_nhSD · 2022-08-09
> > **Thank you for the response!**
> >
> > Thank you for the thoughtful response to my review and others. I will keep my current rating and hope to see this paper accepted. Nice work authors!

---

### Official Review · Reviewer_QHi3 · 2022-07-10

**Rating:** 6
**Confidence:** 3
**Soundness:** 3 good
**Presentation:** 3 good
**Contribution:** 3 good

**Summary:**

In this paper, the authors propose a coded residual transform (CRT) to improve generalization of  deep metric learning method. They use a set of learned anchors to encode the image embeddings. The proposed CRT obtains state-of-the-art performance on four datasets including CUB-200-2011, Cars-196, Standard Online Products and In-shop Clothes Retrieval.

**Questions:**

1. As stated in weakness, the insight of margin-based softmax is similar to CRT. For Eq. (3) and Eq. (4), why not to use softmax formula such as n-pair loss [a] ?
2. The details of batch samplers for training should be clarified in the experiment settings.
3. For the total loss function (Line 205), is MS-loss necessary? Other metric learning loss such as Siamese loss, triplet loss and n-pair loss should be compared.

[a] Sohn, Kihyuk. "Improved deep metric learning with multi-class n-pair loss objective." Advances in neural information processing systems 29 (2016).

**Limitations:**

See weaknesses and Questions.

**Strengths And Weaknesses:**

Strengths:

1. The authors proposed a coded residual transform (CRT) to improve generalization for deep metric learning.
2. The authors utilize a set of earned anchor features to encode the image embeddings. The anchor diversity loss, CRT consistency loss and MS-loss [11] is used for optimization.
3. The experiments and ablation studies are solid.

Weaknesses:

The authors should discuss and compare margin-based softmax [a][b][c] in the sections of  related work and experiments. Although the CRT is different from margin-based softmax, the insight is similar. The parameters W in softmax can be treated as the anchor features.

[a] Liu, Weiyang, Yandong Wen, Zhiding Yu, and Meng Yang. "Large-Margin Softmax Loss for Convolutional Neural Networks." In ICML. 2016.

[b] Wang, Hao, Yitong Wang, Zheng Zhou, Xing Ji, Dihong Gong, Jingchao Zhou, Zhifeng Li, and Wei Liu. "Cosface: Large margin cosine loss for deep face recognition." In Proceedings of the IEEE conference on computer vision and pattern recognition, pp. 5265-5274. 2018.

[c] Deng, Jiankang, Jia Guo, Niannan Xue, and Stefanos Zafeiriou. "Arcface: Additive angular margin loss for deep face recognition." In Proceedings of the IEEE/CVF conference on computer vision and pattern recognition, pp. 4690-4699. 2019.

---

> ### Author Response · Authors · 2022-08-02
> **Response to comments on margin-based softmax,  batch samplers, the total loss function and other metric learning loss.**
>
> We sincerely thank you for this thorough and insightful review of our paper! We also appreciate your positive and encouraging comments about our paper! In the following, we provide detailed response to your comments.
>
> $\textbf{1. Response to comments on margin-based softmax and Eqs. (3) and (4):}$
>
> Thanks for this detailed comment! In the revised paper, we have followed your comment and discussed the margin-based softmax methods in related work. We agree that the parameters W in softmax can be treated as the anchor features. We actually have tried this during our experiments and the performance degradation is very large. The major reason is that the softmax only provides a high-level global description of the input. However, in our paper, the set anchors learned from the training set sever as the building blocks of the image scenes. They correspond to different scene objects at different spatial locations in the input image.  The anchor distributions at the training set and the test set could be different.
>
> We followed your suggestion and test the performance of arcface on the CUB dataset based on the ResNet-50 backbone, the top-1 retrieval accuracy is 61.39%, which much lower the proposed CRT method that has obtained 64.20% top-1 retrieval accuracy under the same experimental conditions.
>
> Sorry for the confusion! We agree that Eq. (3) and (4) are related to softmax. However, they are different. The softmax performs normalization on one vector. Eqs. (3) and (4) defines the correlation between two feature vectors and the similarity matrix for a set of features. We use the cosine similarity. The value of the cosine similarity computed on the L2 normalized features is between 0 and 1, which is similar with the softmax normalization. It should be noted that before computing the correlation, we did perform the softmax-like normalization on the feature.
>
> $\textbf{2. Response to the comment on batch samplers for training in the experiment settings:}$
>
> Thanks for this suggestion! We have added the description of the batch samplers for training in the revised paper (Section 4.1). During training, we randomly sample a set of images in each iteration to train the network. In each iteration, we first randomly choose the image classes, and then randomly sample 5 images from each class to form a batch. For the CUB and Cars datasets, we sampled 16 classes in a batch. For the SOP and In-Shop datasets, we sampled 36 classes in a batch.
>
> $\textbf{3. Response to the comment on the total loss function and other metric learning loss such as Siamese loss, }$
> $\textbf{triplet loss and n-pair loss for comparison: }$
>
> Thanks for this valuable comment! The MS loss function is necessary for the total loss function. First, from our experiments, we observe that the MS loss is required since it is a very effective loss for metric learning. It provides a very important starting point for our proposed method. Without this, the metric learning performance will be degraded.
>
> Another reason is that the MS loss function considers all the similarity relationship between samples in a batch, which is important for the training of a well generalizable model. It is also consistent with other two loss function terms. However, the Siamese loss, triplet loss and n-pair loss functions only consider the similarity between a set of samples with the anchor, which is a small portion of the similarity relationship between samples in the whole batch. For example, the N-pair loss did not consider the similarity relationship in N negative examples.
>
> To verify this phenomenon, we conducted new experiments using the N-pair loss to instead of the MS loss under the same experimental conditions on the CUB dataset. The top-1 retrieval accuracies are 56.1% and 75.9% for the N-pair loss and MS loss, respectively. This result demonstrated the need and advantage of using the MS loss.

---

> > ### Comment · Reviewer_QHi3 · 2022-08-08
> > **Final rating**
> >
> > After reading other reviews and rebuttals, I decide to improve the final rating.

---

> > > ### Author Response · Authors · 2022-08-08
> > > **Thank you very much!**
> > >
> > > Dear Reviewer,
> > >
> > > We  really appreciate your time and efforts in reviewing our paper, and your valuable comments for improve our paper! Thank you very much for deciding to improving the final rating!

---

### Official Review · Reviewer_jY4B · 2022-07-10

**Rating:** 4
**Confidence:** 4
**Soundness:** 2 fair
**Presentation:** 2 fair
**Contribution:** 2 fair

**Summary:**

This work proposes to apply residuals to the anchors as representations for inputs. Compared with the features directly extracted from inputs, the residuals provide the relative information with a learnable codebook, which can alleviate the overfitting problem. Besides, an additional branch is included for the consistency between representations from different codebook.

**Questions:**

My major concerns are about motivation and the fair comparison issue as listed in weakness.

**Limitations:**

Compared baseline methods, the additional branch for consistency constraint may require more computational resources/running time.

**Strengths And Weaknesses:**

Strong
1.	Adopting residuals for generating features is novel for deep metric learning.
2.	The ablation study shows that a consistency constraint is helpful for learning representations.
3.	The experiments is sufficient on benchmark data sets.


Weak
1.	The motivation of introducing residuals for better generalization on unseen classes is not convicting. Note that the anchors are also learned from the training data, which cannot work as reference points for unseen classes. For example, according to the example in Line 57, the anchor points will only contain daytime faces from training while the residuals of night-time faces to those anchors cannot capture the appropriate information.
2.	The additional branch increases the training cost, which should be discussed.
3.	The performance of MS in Table 5 is degenerated compared to the original paper, which reports the similar performance without the proposed method. Since many previous works have BN-Inception as the backbone, it is better to include the results with the same backbone and same dimension of features for a fair comparison.

---

> ### Author Response · Authors · 2022-08-02
> **Response to the comments on anchors, additional branch, the training cost, MS loss and BN-Inception.**
>
> We really appreciate your time and efforts on thorough review of our paper and your valuable comments! Also thank you for your positive and encouraging comments about our paper: $\textit{“Adopting residuals for generating features is novel for deep metric learning…}$$\textit{The experiments is sufficient on benchmark data sets.”}$
>
> In the following, we provide detailed response to your comments. We hope these responses are able to address your concerns.
>
> $\textbf{1. Response to the comment on “anchors are also learned from the training data, which cannot work as reference points}$
> $\textbf{for unseen classes. For example… daytime faces… night-time faces”.}$
>
> Thanks for this very insightful comment! Yes, in our current design, the anchors are learned from the training side and also used for the test unseen classes. According to our analysis, the anchors can be considered as a dictionary of prototypes which construct the image scenes, both in the training and test datasets. Although they share the same anchor set, we observe that the anchor distributions at the training side and the test side are different. This fact is aligned to the existing research on distribution shift.
>
> In our future work, motivated by your comment, we will investigate how the dictionary of anchors can be updated for the unseen test classes to further improve the generalization capability and performance of our method.
>
> $\textbf{2.  Response to the comment on additional branch and the training cost:}$
>
> Thanks for pointing out this! Following your comment, we have added a discussion on the extra training cost caused by the additional branch in the revised paper. The parameters of the backbone network are shared for these two CRT branches. According to our estimation, the increase of training cost is only about 5%. The second branch is only used during training to provide guidance, which will not increase the test cost. Therefore, it will not affect the complexity at the test time.
>
> $\textbf{3. Response to the comment on MS loss and BN-Inception backbone: }$
>
> Thanks for this valuable comment! For the MS method, the results reported in the original paper is based on an embedding size of 512. In our paper, for fair comparison with other methods, we set the embedding size to be 128 as reported in Table 5, since most recent methods reported in the literature are using the dimension of 128. The choice of 128 is important for large-scale datasets and real-time applications. The results of BN-Inception have already been reported in Table 5 in the original paper. Sorry that we did not mention this clearly.

---

> > ### Comment · Reviewer_jY4B · 2022-08-08
> > **Final rating**
> >
> > Thank authors for the response. The rebuttal addresses my second question but that for the others are not satisfied. First, the motivation is inappropriate as confirmed by authors. Second, it is important to apply state-of-the-art configuration for a fair comparison. A better performance with 512 features is more convincing than a comparable performance with 128 features. Therefore, I would like to keep my rating since the work can be further polished. However, it is OK to accept the paper.

---

> > > ### Author Response · Authors · 2022-08-09
> > > **Thank you very much!**
> > >
> > > Dear Reviewer,
> > >
> > > We really appreciate your kind reply!
> > >
> > > Sorry for the confusion about the motivation part. In the original paper, we used the **toy example** of daytime and night time faces to motivate the following thinking: an effective face detection with sufficient generalization power will not focus on the absolute pixel values of the face image. Instead, it detects the face based on the relative change patterns between neighboring regions inside the face image. From the generalization point of view, we find that it is more effective to learn the embedding based on relative difference between features since the absolute value of features may vary significantly from the training to new test classes, but the relative change patterns between features may have less variations.
> > >
> > > During our experiments, we find that, although the training and testing classes share the same project anchors, the projection anchors of the training classes and test classes have different distributions. During our coded residual transform, we assign different weights for different project residuals based on the correlation between the feature and the corresponding anchor. Therefore, for training classes, the subset of anchors which are close to the training images will have larger weights. Similarly, for the testing classes, the subset of anchors which are close to the test images will have larger weights. This correlation-based weighting for the projection residual is the main idea of our method and contribute significantly to the overall performance gain.
> > >
> > > We have revised the paper in the Introduction section. We hope that this has addressed your concern.
> > >
> > > For your original comment 3 on performance comparison with the feature dimension of 512, we have followed the MS paper and conducted experiments using the BN-Inception backbone network on the CUB dataset.  The top-1 recall rate is 66.5% for the proposed CRT method, which is 0.8% higher than the MS method (65.7%). We have added this result to the Supplemental Materials in the revised paper.
> > >
> > > Thank you so much!
> > > Authors

---

### Meta-Review · Area_Chair_yzDh · 2022-08-27

**Recommendation:** Accept
**Confidence:** Less certain

**Metareview:**

This paper proposes a coded residual transform for deep metric learning, aiming to improve the generalization ability of metric learning to unseen classes. Four expert reviewers assessed this paper, with preliminary reviews at odds. After author rebuttal, some reviewers acknowledged the rebuttal by increasing the score, but one reviewer still held major concerns including the vague motivation and unconvincing evaluation. AC read the paper itself as a neutral referee, and considered all reviewing material. AC's take on the paper is as follows.

- The motivation of using coded residual to improve generalization to unseen classes lacks a sufficient elaboration, either theoretical or technological -- showing the motivation with a toy example is not strong. Also, there is no clear connection between the coded residual and the generalization property in Roth et al. [22] -- again, only a partial result in Table 1 is not enough.

- The algorithmic contribution is slightly below the bar of NeurIPS. By toothing apart each component, only Eq. (1) regarding the coded residual is somewhat novel to me. Eq. (2) is a common feature combination method, similar to concat, element-sum, or the aggregation used in GNNs and PointNet. Eq. (3) is a common criterion in linear discriminant analysis. Eq. (4) is something like the kernel matrix, which is of O(N^2) complexity, very time-consuming to compute for deep learning. In all, the technical novelty is relatively slim while the practical value in terms of efficiency is not high.

- Writing is problematic in some way. Authors try to explain the motivation of coded residual but in fact the idea was firstly proposed in the classic method VLAD [49]. Perhaps the new thing is that authors also try to explain that the coded residual is more generalizable to unseen classes, but this is less elaborated. Authors shall clearly credit to VLAD and limit their own contribution to "generalizing to unseen classes".

Nonetheless, AC feels that the idea of coded residual is interesting in the metric learning context, and authors are suggested to work forward for a stronger approach that generalizes learned metrics to unseen classes, which is important for practical open-world applications. After discussion between AC and SAC, their opinion on this paper was somewhat in disagreement. SAC suggested that by taking the reviews as well as the scores (8/7/6/4) into consideration, the paper should be accepted, and the negative points above can be addressed by the authors in the next version. AC revised the metareview from (Borderline) Reject to (Borderline) Accept accordingly.

**Award:**

No

---

### Decision · Program_Chairs · 2022-09-14

Accept